# Integrative Approaches in Non-Small Cell Lung Cancer Management: The Role of Radiotherapy

**DOI:** 10.3390/jcm13154296

**Published:** 2024-07-23

**Authors:** Maxime A. Visa, Mohamed E. Abazeed, Diego Avella Patino

**Affiliations:** 1Department of Surgery, Northwestern University Feinberg School of Medicine, Chicago, IL 60611, USA; maxime.visa@northwestern.edu; 2Department of Radiation Oncology, Northwestern University Feinberg School of Medicine, Chicago, IL 60611, USA; mabazeed@northwestern.edu

**Keywords:** radiotherapy, immunotherapy, NSCLC, chemoradiotherapy, abscopal effect

## Abstract

Treatment guidelines for non-small cell lung cancer (NSCLC) vary by several factors including pathological stage, patient candidacy, and goal of treatment. With many therapeutics and even more combinations available in the NSCLC clinician’s toolkit, a multitude of questions remain unanswered vis-a-vis treatment optimization. While some studies have begun exploring the interplay among the many pillars of NSCLC treatment—surgical resection, radiotherapy, chemotherapy, and immunotherapy—the vast number of combinations and permutations of different therapy modalities in addition to the modulation of each constituent therapy leaves much to be desired in a field that is otherwise rapidly evolving. Given NSCLC’s high incidence and lethality, the experimentation of synergistic benefits that combinatorial treatment may confer presents a ripe target for advancement and increased understanding without the cost and burden of novel drug development. This review introduces, synthesizes, and compares prominent NSCLC therapies, placing emphasis on the interplay among types of therapies and the synergistic benefits some combinatorial therapies have demonstrated over the past several years.

## 1. Introduction

Lung cancer persists as one of the most pervasive oncological challenges, accounting for approximately 2.5 million new cases and 1.8 million deaths worldwide in 2022 alone [1]. As the most common form of lung cancer, non-small cell lung cancer (NSCLC) accounts for 85% of all lung cancer cases, making for a highly prevalent disease with high mortality rates despite advances in treatment [2]. Sixty percent of NSCLC cases are diagnosed at stage III or above when dissemination to lymph nodes and/or distant organs has already occurred, underscoring the challenge that clinicians face [3,4]. NSCLC treatment at early stages is curative in nature and includes surgical resection and ablative therapies directed at the primary tumor. While recent lung cancer screening initiatives have increased the proportion of NSCLCs caught in early stages, NSCLC is infamous for its ability to remain undetected for long periods of time, at which point surgical resection and curative ablative therapies may no longer be feasible [5]. In advanced-stage tumors, a combination of treatments, including chemotherapy, radiotherapy (including conventional fractionation and hypofractionation), molecular targeted therapy, surgical resection, and immune checkpoint blockade (ICI), are often used for NSCLC management [6,7].

Recent advancements in the molecular understanding of NSCLC and its constituent microenvironment have paved the way for modern NSCLC treatment [8]. Overall, NSCLC outcomes in the United States have improved in the past few years, much of which can be directly attributed to early cancer detection as a function of early-screening programs and the development of novel gene targeted therapies like tyrosine kinase inhibitors (TKIs) and immunotherapies targeting programed death-ligand 1 (PD-L1), programed death-1 (PD-1), and cytotoxic T-lymphocyte associated protein-4 (CTLA-4) in the ICI family [7,9]. Complete surgical resection of primary NSCLC is the mainstay of NSCLC treatment in early stages. Surgical resection of the primary tumor for early pathological stage (I-II) NSCLC is the most effective treatment, with a 5-year recurrence-free survival of 80% [10]. However, the minority of patients diagnosed with NSCLC are candidates of surgical resection. In patients for whom surgical intervention is contraindicated, stereotactic body radiation therapy (SBRT) is a highly effective and feasible alternative as definitive therapy with curative intent. In this population with a high rate of co-morbidities, the overall survival at two years can range from 50 to 70% [11,12,13], with a 5-year overall survival of ~40% [13]. Historically, patients with locally advanced NSCLC were treated with a combination of chemotherapy, radiotherapy, and surgical resection administered in different regimens preoperatively or postoperatively. Patients with involvement of a single mediastinal lymph node station (N2) or tumors with invasion of the chest wall or pleura and N1 disease were treated with neoadjuvant chemoradiation followed by surgical resection. Patients that underwent surgical resection had longer progression-free survival in comparison to the patients that received only radiation and chemotherapy [14,15]. The development of ICI therapy has revolutionized the treatment of NSCLC and has made it more complex, requiring a truly multidisciplinary approach for the treatment of the majority of patients with a diagnosis of NSCLC. Indeed, the field has expanded so rapidly and is continuing to grow at such a meteoric pace, making it impossible to cover all aspects of current NSCLC therapy in depth and with appropriate precision. This review will focus on the clinical outcomes related to the use of a combination of therapeutic regimens, particularly related to the addition of radiotherapy to ICI. 

## 2. Methods

Selected articles for this narrative review were obtained through literature searches across PubMed, Scopus, Google Scholar, and Web of Science to maximize catchment of relevant articles. Our search strategy was predicated on the use of keywords “NSCLC”, “combinatorial”, “immunotherapy”, “radiotherapy”, and “surgical resection”. Data extracted included patient demographics, tumor characteristics, treatment outcomes, adverse events, and study limitations. We did not enforce any specific time period in our search, although the majority of selected articles represent novel research in the spirit of relevance. Inclusion criteria included articles that discussed immunotherapy or radiotherapy individually or in tandem for NSCLC of all stages, as well as surgical resection with either immunotherapy or radiotherapy. Exclusion criteria included articles that were not available in full text, studies published in languages other than English, and gray literature, including conference abstracts, dissertations, and non-peer reviewed articles.

While not a systematic review, the structure of our informal literature search represents our best attempt to include as many publications relevant to novel NSCLC treatment as possible. Our pointed selection of articles represent a targeted and effective amalgam that presents trusted, standard-setting, and novel research across several research types, to include clinical and laboratorial research as well as systematic reviews and meta-analyses. This review was structured in a manner consistent with the Scale for the Assessment of Narrative Review Articles (SANRA) [16].

## 3. Rationale for Immunotherapy 

With the advent of ICIs, several studies and trials have demonstrated improved primary endpoints using ICI as first- and second-line treatments for treating late-stage locally advanced and metastatic NSCLC. The benefits of ICI therapy have been replicated in multiple cancers including head and neck cancer, melanoma, bladder cancer, etc. Mechanistically, ICIs unblock the host’s adaptive immune response using monoclonal antibodies tailored toward ligands or receptors with inhibitory motifs like PD-1, PD-L1, and CTLA-4. As such, a more robust host adaptive immune response is observed, leading to improved tumor killing via the enhancement of anti-tumor activity of the immune cell population [17].

The KEYNOTE-024 trial was the first phase-three trial that demonstrated the safety and benefits of ICIs in advanced NSCLC. In this study, researchers assigned 304 recruited patients who had been diagnosed with stage IV NSCLC with a PD-L1 expression of >50% and no sensitizing epithelial growth factor receptor (EGFR) or anaplastic lymphoma kinase (ALK) mutations to either receive pembrolizumab (anti-PD-1 ICI) or platinum-based chemotherapy. Median progression-free survival (PFS) in the treatment arm (pembrolizumab) was 10.3 months versus 6.0 months in the control arm (chemotherapy). At 6 months, 62.1% of patients in the experimental arm were alive and had no disease progression compared to 50.3% in the control arm. With respect to overall survival (OS), 80.2% of patients who had received pembrolizumab were alive at 6 months versus 72.4% in the chemotherapy group. Any grade adverse effects of treatments occurred in 73.4% of patients who had received pembrolizumab versus 90.0% of patients who had received chemotherapy. Together, these results demonstrate a marked improvement in all of the primary endpoints that were assessed in patients who received pembrolizumab versus chemotherapy. The results of this landmark study were pivotal to study the combination of multiple therapies for lung cancer, including radiation, surgical resection, and a combination of ICIs, both preoperatively and postoperatively [18].

## 4. The Role of Radiotherapy in NSCLC

Radiotherapy is widely used across cancer types, with over half of cancer patients receiving treatments during the course of their management [19,20,21]. For early-stage NSCLC, stereotactic body radiotherapy (SBRT) is usually reserved for patients who do not meet the criteria for surgical intervention or those who opt out of surgery. Patients who display a multitude of comorbidities, advanced age, low cardiopulmonary function, or some combination of the three, are typically treated with SBRT [22]. Up to 20% of patients diagnosed with stage I NSCLC do not meet the criteria for surgery [23]. Retrospective studies, however, suggest that SBRT and surgical resection provide equivalent long-term outcomes for NSCLC at early stages [24]. Currently, there are two prospective trials designed to answer this question [25,26]. 

For more advanced-stage NSCLC, radiotherapy was historically part of the definitive treatment in combination with chemotherapy with and without surgical resection [14,15,27]. Therefore, the control arm of the groundbreaking PACIFIC trial used the standard chemoradiation regimen at the time plus placebo and compared it with the experimental group that received standard chemoradiation followed by immunotherapy in the form of ICI. The patients included in this trial were patients with stage III, locally advanced unresectable NSCLC. All patients received two or more cycles of platinum-based chemotherapy in combination with definitive radiotherapy at a dose of 54 to 66 Gy. The experimental group received durvalumab, an anti-PD-L1 monoclonal antibody. The group that received ICI had a longer progression-free survival. These results demonstrated the synergistic effect of ICI with the standard of care at the time that included chemotherapy and radiotherapy [28]. The KEYNOTE-001 trial demonstrated prolonged progression-free survival in patients with advanced NSCLC treated with pembrolizumab, an anti PD-1 monoclonal antibody, in comparison with the group treated with chemotherapy or chemoradiation or who were treatment-naive. A subsequent secondary analysis of this cohort of patients demonstrated that patients who received any form of radiotherapy before pembrolizumab had a longer disease-free survival when compared with patients who received only chemotherapy or were treatment-naive before pembrolizumab [29]. Interestingly, in both trials mentioned above, the results were not affected by the level of expression of PD-L1. 

To date, the study that offers more insights related to the benefits and mechanisms of anti-tumor effects of combining immunotherapy with radiation is the PEMBRO-RT study [30]. This study enrolled 76 patients with metastatic NSCLC to explore the outcomes associated with pembrolizumab administration (200 mg/kg every 3 weeks) following SBRT at the primary tumor site (three doses of 8 Gy) versus pembrolizumab alone. The overall response rate (ORR) at 12 weeks was 18% in the control (pembrolizumab only) arm versus 36% (radiotherapy plus pembrolizumab) in the experimental arm. PFS was 1.9 months in the control arm and 6.6 months in the experimental arm, although this difference was not significant. A median OS of 7.6 months was observed in the control arm, compared to 15.9 months in the experimental arm, although this difference was not statistically significant either. However, the improved ORR, PFS, and OS in patients who received pembrolizumab after SBRT constitute an incipient argument for the additive effects of immunotherapy and radiotherapy. This phenomenon is supported by an MDACC trial, which explored the clinical outcomes associated with pembrolizumab administration (200 mg every 3 weeks) with or without concurrent radiotherapy (SBRT 50 Gy in 4 fractions or traditionally fractionated radiotherapy 45Gy in 15 fractions) regardless of tumor PD-L1 expression in 100 patients with metastatic NSCLC. The ORR found in the control arm was 25% versus 22% in the experimental arm. The median PFS of the control arm was 5.1 months compared to 9.1 months in the experimental group. While the ORR results in this trial conflict with the results of the PEMBRO-RT trial, the PFS findings concur with similar work. The most relevant findings from these trials were that when stratified by PD-L1 status, inter-group results vary drastically. Filtered by high PD-L1 expression (>50%), ORRs were 22% and 25% and PFS was 20.6 and 5.6 months in the control and experimental arms, respectively. However, in low PD-L1 expression tumors (<49%), ORRs were 0% in the control arm and 33% in the experimental arm, and corresponding PFS times were 4.6 and 20.8 months. In PD-L1 negative tumors, ORRs were 30% and 11% in the control and experimental arms, respectively, with corresponding PFS times of 14.2 and 7.8 months. While this study found no significant differences in ORRs or PFS time, the results imply that patients with tumors containing low PD-L1 expression would benefit from a combination of pembrolizumab plus radiotherapy treatment, while those with high PD-L1 expression or PD-L1-negative tumors may not. The effects of pembrolizumab on PFS in high PD-L1 tumors are similar to the effects that pembrolizumab plus radiotherapy have on PFS in patients with low PD-L1 tumors. These data suggest that the immunogenic effects of radiation, and thus use of radiotherapy, might be determined by the expression of PD-L1. However, other variables such as the optimal type, dose, and timing of radiation to enhance the immunologic effects of ICI need further exploration. Also, the role of the type of ICI and the timing of ICI in relation to the timing of radiation of the primary or metastatic tumors deserve further consideration. The increased evidence of the additive positive effects of radiation and ICI has fueled current trials designed to demonstrate the effects of SBRT and ICI in early stages of NSCLC, including in the PACIFIC-4, SWOG 1914, and Keynote 867 trials [31,32,33]. 

The treatment of non-resectable stage I NSCLC was conventional radiotherapy until more recent studies demonstrated that other radiotherapy modalities may offer better outcomes. Li et al. demonstrated in a systematic review and meta-analysis that stereotactic body radiotherapy (SBRT) is superior to conventional radiotherapy (CRT) in treating stage I NSCLC [34]. Of the fourteen out of seventeen total studies analyzed that focused on overall survival, all of them found that 1-, 2-, 3-, 4-, and 5-year OS rates were higher in the SBRT groups (86.23%, 69.26%, 54.73%, 40.36%, 29.30%) than the CRT groups (77.80%, 53.76%, 39.50%, 27.47%, 27.47%). Further, in five articles that evaluated lung cancer-specific survival (LCSS), seven articles that evaluated local control rate (LCR), and four articles that compared progression-free survival (PFS), SBRT was significantly superior to CRT in all metrics. Of the six articles that quantified adverse effects (AE), a significant reduction in dyspnea (RR = 0.77), radiation pneumonitis (RR = 0.52) and esophagitis (RR = 0.30) was observed following SBRT in comparison with CRT with no significant differences in the incidence of other AEs. An MDACC trial supports Li et al.’s findings that SBRT is superior to CRT, demonstrating higher anti-tumor effects in tumors outside of the field of radiation (abscopal effect), with better overall response rates (ORRs) (33%) in patients who received SBRT compared to patients who received traditional radiotherapy (17%) [35].

Although many questions persist depending on what concurrent therapies are administered and the mechanisms by which hyper-fractionated and hypo-fractionated doses function, radiotherapy dose and fractionation schedules in the treatment of NSCLC have been relatively well investigated. Comparing radiotherapy studies with one another is made more difficult by the numerous inter-study variability among modulatory radiotherapy variables including frequency of dose administration, treatment duration, dose fractionation, total dose, radiotherapy type (CRT vs. SBRT), and whether any concurrent therapies (i.e., ICI, chemotherapy) were administered. Differences in these parameters make it increasingly difficult to ascertain what is responsible for the differences observed between studies. A review that compared the results of studies with hyper-fractionation, hypo-fractionation, moderate hypofractionation with or without chemotherapy, and total dose in radiotherapy concluded that there was not sufficient evidence that dose intensification or escalation in any direction had any significant effect on outcomes [36]. However, not all radiotherapy-optimizing studies are inconclusive; one study that compared a 2.75 Gy hypo-fractionated regimen against a 2 Gy conventional regimen found that the latter was associated with a longer median survival of 29 months versus the former’s median survival of 25 months [37]. The RTOG 0617 trial demonstrated that a 60 Gy fractionated dose (in 2 Gy fractions) is superior to a larger 74 Gy dose [38]. In summary, radiotherapy has been demonstrated to have important therapeutic effects in treating NSCLC independent of modulating variables, which can be further amplified with sequential immunotherapy [39].

## 5. Mechanistic Approach of Radiation-Enhanced Anti-Tumor Effects of ICI

Recent developments in our understanding of radiotherapy have elucidated possible immunomodulatory mechanisms by which radiotherapy functions beyond the cellular death and senescence caused by direct treatments. The abscopal effect describes the vaccine-like properties that localized radiotherapeutic treatments have purported to have on the systemic immune response to tumors beyond the lesion ablated [40]. Scientific evidence, however, has produced contradictory results in terms of the ideal type or dose of radiation. These anti-tumor-mediated immune effects of radiotherapy are more frequently associated with low doses [41,42]. 

Unlike the tumoricidal dose of radiotherapy, low-dose radiotherapy can reactivate the tumor microenvironment by mobilizing innate and adaptive immunity [43,44]. In contrast, high doses of radiation may induce immunosuppressed effects in the tumor microenvironment mediated by Treg cells and tumor-associated macrophages. However, there is conflicting experimental data related to the effects of low- and high-dose radiation in the cellular composition of the tumor microenvironment. Myeloid-derived suppressor cells (MDSCs) are known to have a tumor-protective effect and promote tumor progression. High doses of radiation induce recruitment and infiltration of tumors and by MDSCs, as well as an increase in the number of MDSCs infiltrating the spleen, lung, and peripheral blood [45]. Other studies have demonstrated a reduction in the number of MDSCs 1 to 2 weeks after high-dose radiation of the primary tumor [46]. Similarly, dendritic cells (DCs) accumulation and function can be affected by radiation [47]. Both immunosuppressor and immunostimulatory effects of both high and low doses of radiation have been demonstrated in experimental cancer models. 

It is well accepted that the in situ vaccination effect induced by radiation is the key mechanism inducing systemic effects after local radiation of the tumor. The death of tumor cells after radiation releases multiple neoantigens and damage-associated molecular patterns (DAMPs), leading to increased antigen presentation through DCs with subsequent DCs maturation and activation and trafficking to tumors of T cells and NKs cells that destroy the tumor tissues [48,49,50]. The understanding of the systemic molecular response to radiotherapy is less clear. An increase in TNFα, IL-1, IL-6, and IL-8 at the site of primary radiation has been demonstrated [51]. However, the systemic and abscopal molecular changes in distant tumors are unknown. 

Additionally, low-dose radiation can directly induce DNA damage in the immune cells that triggers a danger signal that induces an inflammatory response as opposed to the lethal effect that a high dose of radiotherapy has on the immune cells [47]. Further research on this topic is needed given the variable degrees of sensitivities to radiation in the different populations of immune cells. 

In experimental murine models for NSCLC, low-dose radiation increased macrophage polarization toward M1 and enhanced the natural killer cells with consequential anti-tumor effects [52,53]. Furthermore, combinations of low-dose radiation and ICI have demonstrated increased immune cell infiltration of tumors and tumor control [41,44,54,55].

In conclusion, low-dose radiation has an immunomodulatory effect that can potentially enhance the anti-tumor effects of ICI.

## 6. Markers of Response

Some biomarkers have been used to predict the response to ICI, including PD-L1 expression, tumor mutation burden, deficient DNA repair markers, and intestinal microbiota [8,56,57,58]. To date, however, there are no markers that predict the response to radiotherapy and the response to combined radiotherapy and ICI. 

Absolute lymphocyte count has been suggested as a marker of higher and more durable responses to ICI [59]. The absolute lymphocyte counts before and after radiotherapy and ICI were associated with PFS and the occurrence of the abscopal effect [60,61]. These findings were corroborated by the demonstration of the association between the occurrence of the abscopal effect and the absolute lymphocyte counts before radiotherapy [60]. The absolute lymphocyte count after radiotherapy was ten times higher in patients that demonstrated an abscopal effect. 

The mechanistic role of the DAMP leading to an increase in the antigen presentation after radiotherapy-induced cell death suggests that DAMPs can be used as markers of the increased immunogenicity after radiotherapy. Calreticulin is one of the most widely studied DAMPs [62,63]. 

## 7. Surgical Resection for NSCLC

Surgical intervention, when possible, offers patients with NSCLC the best chance of survival. This may be in part due to the fact that surgery is typically indicated for cancers in their early stages, and patients with several comorbidities are often not good candidates for surgical resection. A review comparing surgical (wedge resection or segmentectomy) versus ablative management of stage I NSCLC demonstrated that among eight criteria-meeting studies, the one-year and two-year pooled PFSs for surgical intervention were 94% and 82%, respectively, both of which were superior to ablative therapies’ one- and two-year PFSs at 86% and 66%, respectively [56]. While not statistically significant, the five-year PFS was 30% for surgery versus 37% for ablative therapy. However, retrospective data including patients who have a longer follow up demonstrated that OS and cancer-free survival at 3 years was superior for patients who underwent surgical resection in comparison to SBRT [64]. Previous randomized trials have failed to answer these questions due to premature closure because of low accrual. As mentioned above, at least two randomized trials designed to answer this important question are currently ongoing. On the other hand, for the most advanced cancers, surgical resection when feasible remains a reasonable therapy. A review comparing treatment options for individuals with stage IV NSCLC found that the 5-year survival rate in individuals who had a surgical resection performed was 25.1%, which was markedly improved compared to patients who received chemoradiation (5.8%), chemotherapy (5.8%), and radiation alone (3.2%) [65]. 

Lung cancer recurrence following surgical resection is significant, with reported rates as high as 68% [66]. In patients with stage IIIa NSCLC, radiotherapy preceding surgical resections boasted improved overall survival (OS) compared with patients who underwent surgical resection alone [67]. While other studies confirm the benefits of administering preoperative radiotherapy alone or with chemotherapy, the specific mechanism by which preoperative radiotherapy improves OS has not yet been elucidated [38,68].

Several trials and studies have recently demonstrated the potential use of neoadjuvant immunotherapy prior to surgical resection for the treatment of NSCLC. Aforementioned metrics including OS and PFS are not present in most of these studies due to the novelty and recency of most of these trials. Instead, primary endpoints include major pathological response (MPR), defined as having <10% viable tumor cells in the resected specimen, as well as pathologic complete response (PCR), in which the resected specimen contains 0% viable tumor cells. Chaft et al.’s interim analysis on stage IB–IIIB resected NSCLC patients who received two doses of neoadjuvant atezolizumab prior to resection reported an MPR of 20% out of 143 patients enrolled [69]. In a similar single-arm study, Gao et al.’s trial enrolled 40 patients to evaluate the MPR and PCR in NSCLC stages IA–IIIB who underwent resection preceded by neoadjuvant sintilimab (PD-1 inhibitor) [70]. That study found that sintilimab led to an MPR in 40.5% of patients with 10.8% patients exhibiting PCR. While both of these studies are limited by their lack of control arms, the high MPRs and PCR in Gao et al.’s results are promising and support the use of an immunotherapy neoadjuvant as opposed to surgical resection.

The CheckMate-816 trial demonstrated that a combination of three cycles of preoperative chemotherapy with ICI in the form of nivolumab, an anti-PD-1 monoclonal antibody, in patients with NSCLC stages IB to IIIA was superior to chemotherapy alone. The group that received ICI had a 37% reduction in the risk of recurrence or death secondary to cancer. Also, 24% of patients who received ICI had CPR in comparison to 2% in the group that received chemotherapy alone [71]. The NADIM trial was a single-arm phase-two trial that evaluated the safety and efficacy of nivolumab and chemotherapy versus chemotherapy alone before surgical resection in 86 patients with resectable stage III NSCLC. In the experimental, using nivolumab plus chemotherapy, arm of this trial, 37% of patients demonstrated a PCR compared to 7% in the chemotherapy-alone group. Further, MPR rates were greater in the experimental group (53%) than in the control group (14%). The NADIM trial also found that patients with tumors expressing PD-L1 > 1% experienced greater benefit from therapy than patients with tumors expressing < 1% PD-L1 [72].

In addition to single-agent immunotherapy preceding surgical resection, several studies have explored the feasibility of preoperative multi-agent neoadjuvant immunotherapy. The NEOSTAR trial is a randomized phase-two clinical trial that compared the combination of three cycles of neoadjuvant nivolumab plus ipilimumab to nivolumab alone before surgical resection of NSCLC stages I to IIIA [73], and it was found that the former was substantially more effective than the latter. The trial found that 22% of patients in the nivolumab arm met the MPR primary endpoint, whereas 38% of patients in the combinatorial therapy (nivolumab and ipilimumab) arm met the MPR endpoint. Further, 9% of patients in the nivolumab group demonstrated PCR, while 29% of patients in the combinatorial therapy group demonstrated PCR. While this study has a small sample size, with only 39 of the 44 patients enrolled undergoing curative surgery, it demonstrated that combination neoadjuvant immunotherapy is associated with better outcomes than single-agent neoadjuvant immunotherapy.

The NeoCOAST trial supported this finding, where trial arms that administered several immunotherapeutic agents had improved endpoints as compared with the single-agent control [74]. Specifically, of the 84 patients enrolled in one of the following arms—durvalumab alone (anti-PD-L1), durvalumab plus oleclumab (anti-CD73), durvalumab plus monolizumab (anti-NKG2A), and durvalumab plus danvatirsen (anti-STAT3 ASO)—the MPR rates in the resected populations were 12.5%, 22.2%, 33.3%, and 33.3%, respectively. The respective PCR rates for each group were 3.7% in the monotherapy arm, 9.5% in the durvalumab plus oleclumab arm, 10.0% in the durvalumab plus monolizumab arm, and 12.5% in the durvalumab plus danvatirsen arm. While these findings further support the use of a combination of neoadjuvant immunotherapy, this study also highlights the fact that some immunotherapy combinations are more effective than others, and further research is necessary to elucidate the combinations that are most effective.

In these trials, a higher percentage of patients in the group that received ICI underwent definitive surgical resection in comparison with patients who received chemotherapy. However, in the CheckMate-816 and the NEOSTAR trials, the main reason for cancelation of surgery was progression of disease from an operable to an inoperable stage, followed by an important percentage of patients who developed significant side effects that precluded them from undergoing surgery. Additionally, the benefit of survival was driven by the antitumor effects in patients with advanced tumors (stage III) as opposed to patients with earlier-stage tumors, which highlights the need to determine which patients would benefit the most from this approach. There are no reliable markers that help guide the selection of patients who would benefit from induction with ICI. Radiographic staging before surgery is not reliable, as demonstrated in the NADIM trial. Of all patients with PCR, 33% and 37% demonstrated stable or partial radiographic response after resection, which indicates that radiographic findings do not correlate with pathologic responses. The NADIM trial obtained blood samples and analysis of circulating DNA before resection. In this trial, positive circulating DNA after induction with ICI was associated with a poorer PFS and OS, but such findings need further validation.

For the purpose of this article, there are no trials to date that compare the role of ICI and chemotherapy with and without RT before surgical resection of NSCLC. However, a single-center study at Cornell evaluated ICI with nivolumab versus nivolumab and SBRT before resection of NSCLC. In 60 patients with resectable stage I–IIIA NSCLC, a randomized phase-two study (N  =  30 each arm) evaluated the addition of two cycles of neoadjuvant durvalumab to stereotactic body radiotherapy (SBRT) versus neoadjuvant durvalumab alone. The experimental regimen was well tolerated compared to the control and demonstrated a higher MPR rate compared to durvalumab alone (53.3% vs. 6.7%) with a crude odds ratio of 16.0 (*p*  <  0.0001). Fifty percent of the tumors that exhibit MPR had PCR [75]. These results are encouraging and support the synergistic effect of ICI and SBRT without chemotherapy; however, validation at a multi-institutional level and long-term survival data are still needed.

The applicability of the findings obtained in these trials outside of high-volume centers is a subject of debate. A multidisciplinary approach to each patient considered for induction therapy followed by an operation should be considered, and ideally, patients should have a team dedicated to thoracic malignancies to provide the most up to date regimens to patients. Surgical resection of peripheral lesions or lesions with small hilar or mediastinal lymph nodes after induction therapy with ICI is not very different than resection in patients that have not had induction therapy and therefore do not represent a surgical challenge. On the other hand, surgical resection of central tumors or tumors with a high lymph node burden are often challenging, necessitating more complex resections and reconstructions justifying surgical resection at centers with more experience in complex lung resections.

## 8. Discussion and Areas Needing Further Investigation

Since the introduction of ICI as part of the treatment for NSCLC, the overall survival has improved. Despite these advances, however, overall survival is still poor. The majority of patients with NSCLC are older and have multiple morbid conditions in addition to cancer, which makes them particularly sensitive to the side effects of therapies. For this reason, patient-reported outcomes are an important measurement that should be considered in addition to the survival and pathologic response to therapy. The three-year assessment of patient-reported outcomes of the CheckMate 001 trial demonstrated that patients who received ICI in the form of nivolumab and ipilimumab had improved global quality of health over time after 30 weeks of treatment, and their mean scores were similar to the general population. Also, patients who received ICI had improved cancer-related symptoms in comparison to chemotherapy [76]. These findings have been reported in other trials of ICI with or without chemotherapy [18,77,78]. There are a few areas that need improvement in the field of patient-reported outcomes, such as the standardization of the tools used to measure the patient-reported outcomes and the validation of these tools to assess the repercussions of ICI toxicities. Effects of the combination of multiple therapies on patient-reported outcomes has not been well established, particularly related to the effects of radiation or surgery in addition to ICI.

The optimal timing of the different components of the therapy, type of immunotherapy, type and dose of radiation, and the type and time of surgical resection are not completely established. One of the therapies that allows for the greatest modulation is radiotherapy, with important variables like dosage, fractionation, schedule, and type of radiation. Ideally, the optimal radiotherapy regimen should maximize tumor ablation and downstream host immunity to tumor antigens while minimizing toxicity. As discussed, it is not immediately evident what radiotherapy dose, fractionation, and/or schedule is associated with the best patient outcomes. However, some preclinical studies have shown that certain ablative parameters may be more effective than others. For instance, Lugade et al. demonstrated that single-dose (15 Gy) radiotherapy resulted in significantly increased immunogenicity and local tumor ablation as compared with a fractionated radiotherapy dose (5 × 3 Gy) in animal models [79]. While these data are promising, such results have yet to be realized for NSCLC in human models. In the context of joint immunotherapy plus radiotherapy, the best radiotherapy regimen has not been well evaluated, as highlighted by Dewan et al. in patients with melanoma [80]. In this study, which combined three different radiotherapy doses (1 × 20 Gy; 3 × 8 Gy; 5 × 6 Gy) with anti-CTLA-4 ICI, only the fractionated radiotherapy regimens (3 × 8 Gy and 5 × 6 Gy) resulted in significant primary (irradiated) and secondary (non-irradiated) tumor inhibition, whereas ICI administration alone had no effect on primary or secondary tumors, and radiotherapy alone affected only the primary tumors. The findings of this study highlight two concepts that are important to consider. First, the optimal radiotherapy regimen when administered alone may not confer the greatest survival benefits when paired with another therapeutic modality; this is made evident by the discrepancies between Lugade et al.’s results, which showed that single-dose radiotherapy was superior to a fractionated dose, and Dewan et al.’s results, which showed that only fractionated doses elucidate the abscopal effect when paired with anti-CTLA-4 ICI. Second, combination immunotherapy and radiotherapy offer synergistic benefits, i.e., immunotherapy and radiotherapy together confer benefits than neither can achieve alone: more effective destruction of secondary tumors outside of the irradiation field.

Studies have demonstrated that the host immunogenic response following radiotherapy ablation varies from one metastatic lesion to another within the same host [81]. Thus, radiotherapies should be targeted at tumor lesions that will sensitize the tumor immune response the best to optimize concurrent immunotherapy and increase the chance of an observed abscopal effect. Poleszczuk et al. proposed a model-based immunogenicity index that emulates T cell trafficking in order to discern which lesion sites might potentiate the greatest immune response based on the anatomic distribution of metastatic sites, volume of each metastasis, and site of activation [81]. In a study assessing concurrent versus sequential ipilimumab following SBRT, Tang et al. confirmed that the metastatic lesion chosen for radiotherapy treatment matters from an immunologic standpoint [82]. Tang et al. found that when liver metastases were irradiated, CD8+ cell immune-marker profiles were different than the CD8+ immune-marker profiles when lung metastases were irradiated. While determining optimal radiotherapy sites will require further work, this could be bypassed via multi-site radiotherapy. Studies have demonstrated improved outcomes in primary endpoints when patients are treated with multi-site radiotherapy compared to local radiotherapy and noted acceptable toxicity profiles in the context of joint multi-site radiotherapy plus ICI administration [83,84].

Like radiotherapy modalities, clinicians are able to choose from several ICIs to treat NSCLC. Popular ICIs include anti-PD-1, anti-PD-L1, and anti-CTLA4 agents; however, it is important to determine which mechanism confers maximum tumor killing when administered in tandem with radiotherapy. Chen et al. sought to determine whether anti-CTLA4 or anti-PD-1 agents offer the best survival metrics when paired with SBRT for NSCLC and found intriguing results. Of 33 patients enrolled in their study, global response rates in the anti-CTLA4 group were 24% versus 56% in the anti-PD1 group. Further, PFS rates for anti-CTLA4 and anti-PD-1 were 23% and 63% at 18 months, respectively [85]. Corresponding OS rates for anti-CTLA4 and anti-PD-1 were 39% versus 66% at 18 months. These results demonstrate that anti-PD-1 agents may provide superior survival benefits when paired with SBRT as opposed to anti-CTLA4. Further evidence regarding treatment-related toxicity supports the use of anti-PD-1/PD-L1 agents opposed to anti-CTLA4 agents. A systematic review and meta-analysis performed by Abdel-Rahman et al. showed that the odds ratio for treatment-related death with anti-CTLA4 (ipilimumab and tremelimumab) was 1.80, while the corresponding odds ratio for anti-PD-1/PD-L1 inhibitor was 0.63 [86]. It should be noted that the studies included in this review did not exclusively include combinatorial immunotherapy plus radiotherapy treatments and included monotherapy ICI trials as well. However, a repeat analysis of the odds ratio for treatment-related death with anti-CTLA4 after the reviewers excluded the only study that used joint immunotherapy plus radiotherapy found an increased odds ratio of 2.95, suggesting that anti-CTLA4-related AEs may be lower when concurrently administered with radiotherapy.

The final important consideration with regard to combinatorial treatment is the order in which immunotherapy and radiotherapy are delivered. Some hypothesize that delivering radiotherapy first allows for increased available tumor antigens resulting from ablation, thereby bolstering the immune response as these tumor antigens are necessary for proper T cell activation. Alternatively, others hypothesize that stimulation of the immune system prior to radiotherapy can result in a superior antigen-presenting cell (APC) response at the time of tumor ablation due to pre-ablative immune sensitization. One study’s results purport that relative timing of immunotherapy and radiotherapy is dependent on the immunotherapeutic agent administered, implying that the different mechanisms by which immunotherapies work may necessitate administration at different times with respect to radiotherapy [87]. Thus, the appropriate clinical decision regarding how to time immunotherapy and radiotherapy may be agent-dependent, and the mechanism of the immunotherapeutic agent should be taken into consideration first and foremost [88].

While the gold standard for NSCLC treatment continues to evolve, it is becoming increasingly evident that treatment optimization will likely involve the combination of several treatment modalities (Table 1). While the rapidly progressing nature of NSCLC treatment has good implications from a scientific standpoint, this comes at the expense of optimized, widely accepted, and tried-and-true guidelines for clinicians to be equipped with. This lack of cemented guidance and standardized protocols, while an opportunity to test treatment efficacy and optimization in clinical trials, is an important limitation in NSCLC treatment today that affects how the disease is approached and tackled from institution to institution and around the world.

Financial barriers must also be considered for multi-therapy regimens, particularly for patients who are poorly are not insured or live in areas of the world where a greater proportion of treatment cost is the patient’s responsibility. While NSCLC treatments vary in cost, no surgical, radioablative, chemotherapeutic or immunotherapeutic agent is inexpensive. Further, the combination of these agents and procedures—while beneficial medically—can substantially increase the cost of treatment [89,90]. For patients who are uninsured or who are paying out of pocket, combinatorial treatment strategies may not be feasible despite their efficacy. The root cause of this financial burden stems directly from the rapidly evolving nature of the field and therapies that we so often praise; heavy research and development costs, cutting-edge technologies, and heavy investment by third parties all work to drive up consumer cost, which is another factor clinicians should be considering when discussing treatment strategies with their patients. Another important limitation to combinatorial therapy is that of patient access. There are important geographical disparities vis-a-vis the availability of specialized medical centers that are able to (i) diagnose NSCLC in a timely fashion and (ii) have the capability to deliver the complex requisite treatments to combat this rapidly progressive and devastating disease. These disparities exist and must be considered not only on a local level, i.e., state- and nation-wide, but also globally. The resources required to adequately deliver NSCLC treatment, let alone combinatorial treatment, can be few and far between and not immediately accessible to patients in middle- and low-income countries, providing a substantial limitation to the implementation of the science we discuss in this review.

In addition to the scientific, logistic, and pragmatic limitations of combinatorial treatment, our review holds limitations as well. First, our selection of relevant and impactful studies aimed at portraying the salient findings and impressions of NSCLC treatment is less geared towards holistically describing the state of combinatorial treatment but more so focuses on the important studies that have shaped the field in the past two decades. As such, the inclusion/exclusion criteria for selected articles and studies were not as stringent as those of systematic reviews, and not all studies that exist and are available on this topic were included in our discussion. As such, our narrative review is open to several biases that are worth mentioning that may otherwise be mitigated in the systematic review process, including, but not limited to, selection bias, as aforementioned, publication bias, citation bias, confirmation bias, subjectivity, and lack of reproducibility. We aimed to mitigate some of these biases by having more than one researcher design, perform, and contribute to our search, by including studies that both confirm and refute common hypotheses, and by describing our search criteria—although a narrative and not systematic review—in the Methods Section.

Another important limitation of this review is the quality of each study we include in our discussion. This review encompasses a wide berth of various study types, including randomized control trials, observational studies, and retrospective reviews, which are held to different standards, which might affect the validity of comparing them to each other as we do here. Further, heterogeneities in study design, outcome measures, patient populations, surgical techniques, radiotherapy dosing, fractionation, scheduling, and immune- and chemotherapy dosing, as well as innumerable other metrics, make it increasingly difficult to draw parallels and conclusions from one study to another. This limitation highlights the importance of clinical trials that directly compare therapy options, such as the CheckMate, NeoCOAST, NEOSTAR, NADIM, and other trials we describe in this review.

## 9. Future Directions

Ultimately, the optimal treatment regimen for NSCLC depends on several factors including tumor histology, staging, and PDL-1 expression, among others. In the area of ICI immunotherapy, there is increasing interest in elucidating the best combination of therapies for each patient. In patients with unresectable locally advanced NSCLC, chemoradiation and immunotherapy are typically administered sequentially and represent the core of modern anti-cancer therapy for this category of patients. The role and scope of radiotherapy in the NSCLC environment is still growing and has been demonstrated to enhance the clinical benefits conferred by immunotherapy. The abscopal effect is a particularly compelling phenomenon that may provide more insight into these putative interactions by which coexisting tumors outside of the irradiation field respond to radiative treatment in some patients with metastatic disease. As shown in the PEMBRO-RT study, a significant clinical benefit was found in patients with metastatic NSCLC who had a PDL-1 expression > 0 yet < 49% that received combinatorial immunotherapy and radiation to the primary tumor. In this tumor demographic, these findings corroborate some interaction between radiotherapy and immunotherapy. Further evaluation of ICI-response biomarkers in addition to PD-L1 expression, such as tumor microsatellite instability, tumor mutation burden, and circulating DNA, will help to determine which patients may benefit the most from this approach. The ideal radiotherapy fractionation, dose, frequency, and relative order of administration to ICI that confers the greatest anti-tumor response is still unknown and necessitates robust future experimentation.

Above, we discuss several potential strategies upon which our field can continue to build. For instance, discrepancies in optimal radiotherapy strategy in the setting of joint immunotherapy plus radiotherapy necessitates studies that aim to optimize radiotherapy dose, fractionation, and schedule with ICI. On the other hand, Chen et al.’s study revealed that future studies must continue to seek optimization of the ICI mechanism and agent against a constant radiotherapy regimen. Further mechanistic work exploring the likes of the abscopal effect can further elucidate how to best select radiotherapy sites and how to maximize the immunogenic effects radiotherapy portends. Last, studies that aim to ascertain the optimal order in which combinatorial therapies are delivered would have important clinical applications and potentially provide further insight into the mechanisms that confer the synergy between radiotherapy and immunotherapy that to date have only been empirically observed.

A limitation to exploring the combination of currently available therapies for NSCLC is the lack of reliable preclinical models. The frequency in which the abscopal effect is seen in murine models and in patients is dramatically different, suggesting that the biology of the murine NSCLC is different as compared with the human NSCLC. The development of large human tumor banks and preclinical models such as organoids and patient-derived xenografts may help to answer these questions in the future.

While rapidly evolving, the anti-cancer effects and clinical benefits of combinatorial ICI, gene-targeted therapy, and radiotherapy remain largely unexplored. By prolonging life and reducing the tumor burden in patients with initially unresectable NSCLC, it is possible that after sufficient treatment, some unresectable tumors could be sufficiently down staged to allow for surgical resection. Adding radiotherapy to induction therapy in locally advanced tumors may improve survival and increase the rates of local control.

## Figures and Tables

**Table 1 jcm-13-04296-t001:** Summary of combinatorial therapy studies for NSCLC, including various combinations of chemotherapeutic and immunotherapeutic agents along radioablative regimens and/or surgical resection.

Study	Arm 1	Arm 2	Arm 3	Arm 4	Stage	Participants	Main Findings
RTOG 0617 [38]	Standard Dose radiotherapy + chemotherapy +/− cetuximab	High-dose Radiotherapy + chemotherapy +/− cetuximab	None	None	IIIA, IIIB	544	Median OS for the standard dose arm was 28.7 months and 20.3 months in the high-dose group. The addition of concurrent cetuximab did not improve survival.
PEMBRO-RT [30]	Pembrolizumab	SBRT + Pembrolizumab	None	None	All	76	ORR at 12 weeks was 18% in the control arm versus 36% in the experimental arm.
MDACC [35]	Pembrolizumab	SBRT + Pembrolizumab	None	None	IV	100	The ORR found in the control arm was 25% versus 22% in the experimental arm. Median PFS of the control arm was 5.1 months compared to 9.1 months in the experimental group.
Chen et al. [67]	Surgical Resection	Radiation + Surgical Resection	None	None	IIIA	2675	Preoperative radiation was associated with better overall survival than surgical resection alone.
Chaft et al. [69]	Atezolizumab + Surgical Resection	None	None	None	1B–IIIB	143	MPR of 20% and a 3-year survival of 80%.
Gao et al. [70]	Sintilimab + Surgical Resection	None	None	None	IA–IIIB	40	Sintilimab led to an MPR in 40.5% of patients, with 10.8% patients exhibiting PCR of the 37 patients that underwent surgery.
NEOSTAR [73]	Nivolumab + Surgical Resection	Nivolumab + Ipilimumab+ Surgical Resection	None	None	I–IIIA	44	More patients in arm 1 met the MPR endpoint and demonstrated PCR than in arm 2.
NeoCOAST [74]	Durvalumab	Durvalumab + oleclumab	Durvalumab + monolizumab	Durvalumab + danvatirsen	IA–IIIA	84	Combinatorial immunotherapy is more effective than single-agent immunotherapy, although some combinations are more effective than others.
NADIM [72]	Chemotherapy + Surgical Resection	Nivolumab + Chemotherapy + Surgical Resection	None	None	III	86	More patients in the experimental arm demonstrated PCR than in the control, and MPR rates were greater in the experimental arm as well.
CheckMate [76]	Chemotherapy + Surgical Resection	Nivolumab + Chemotherapy + Surgical Resection	None	None	IB–IIIA	358	Decreased disease progression, increased MPR rates, and greater proportion of PCR in experimental arm compared to control arm.
Tang et al. [82]	Ipilimumab concurrent to SBRT	Ipilimumab sequential to SBRT	None	None	IV	35	Irradiation of sites of hepatic metastases conferred greater T cell activation than irradiation of lung tumor sites. No difference in concurrent vs. sequential therapy.
Chen et al. [85]	Radiation + ipilimumab	Radiation + pembrolizumab	None	None	N/A	N/A	Retrospective comparison of two studies demonstrated that anti-PD-L1 agents may yield improved survival metrics compared to anti-CTLA-4 agents when combined with radiation.

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
