# Peer review of "Integrative Approaches in Non-Small Cell Lung Cancer Management: The Role of Radiotherapy"

_jcm, 2024, doi:10.3390/jcm13154296_

Round 1

Reviewer 1 Report

Comments and Suggestions for Authors

Thank you for submitting this interesting manuscript to the Journal of Clinical Medicine. I was pleased to receive it as a reviewer.

Your manuscript provides a useful review of the current state and future directions of combinatorial therapies in NSCLC. To further improve the quality and impact of your work, I would like to offer the following suggestions:

1.      Consider including a Methods section to increase the transparency and credibility of your review. I understand that this is not a systematic review; however, you could mention the databases you searched, some keywords, and your inclusion/exclusion criteria. If you're interested, check out the Scale for the Assessment of Narrative Review Articles (SANRA; doi: 10.1186/s41073-019-0064-8) for some guidance.

2.      Consider providing a diagram or a table to summarize the key combinatorial therapies discussed in the manuscript. This would help readers quickly grasp the main concepts without getting lost in the details.

3.      Consider expanding the discussion on the challenges of implementing combinatorial therapies in clinical practice (e.g., financial considerations, patient access, need for standardized protocols). This addition would make your review more relevant to a wider audience, including researchers and policymakers.

4.      In the era of precision medicine, biomarkers are all the rage. You may consider including a section on potential biomarkers that could guide patient selection for combinatorial therapies in NSCLC (e.g., PD-L1 expression, tumour mutational burden, specific genetic alterations). I would also suggest expanding your discussion on the potential role of circulating tumour cells and cell-free DNA in guiding combinatorial therapies. Blood-based biomarkers are a hot topic right now.

5.      In section 4 (Surgical Resection for NSCLC), consider discussing the percentage of patients who did not proceed to surgery after neoadjuvant immunotherapy and the intraoperative complication rates. It's also worth considering whether the impressive surgical outcomes reported in relevant randomized trials can be replicated in lower-volume centres. What are your thoughts on this?

6.      You've done a great job discussing the clinical evidence for combinatorial therapies in NSCLC, but I think you can take it a step further. Can you provide more detail on the underlying biological mechanisms that make these combinatorial therapies work so well together? For example, what specific immune cell populations and molecular pathways are involved in creating these synergistic effects?

7.      Have you thought about including a section on quality of life and patient-reported outcomes? It's important to consider how these combinatorial therapies affect patients beyond just their clinical outcomes. Are these approaches harder for patients to tolerate compared to conventional therapy? What do NSCLC patients actually prefer?

8.      Consider discussing in detail the limitations of your review. It's important to be transparent, both in terms of the inherent limitations of a narrative review and any specific limitations related to your manuscript (e.g., quality of included studies).

I hope these suggestions are helpful, and I wish you the best in the publication process.

Author Response

Reviewer 1:

  1. 1. Consider including a Methods section to increase the transparency and credibility of your review. I understand that this is not a systematic review; however, you could mention the databases you searched, some keywords, and your inclusion/exclusion criteria. If you're interested, check out the Scale for the Assessment of Narrative Review Articles (SANRA; doi: 10.1186/s41073-019-0064-8) for some guidance.

            We thank the reviewer for suggesting to add a methods section to our manuscript for transparency. While not a systematic review, we believe that our manuscript’s credibility is much improved, and the replicability of our narrative review—while never perfect—is now more feasible. Our methods section, found on lines 89 – 107, is italicized below for ease of reading:

“Selected articles for this narrative review were obtained through literature searches across PubMed, Scopus, Google Scholar, and Web of Science to maximize catchment of relevant articles. Our search strategy was predicated on the use of keywords “NSCLC”, “combinatorial”, “immunotherapy”, “radiotherapy”, and “surgical resection”. Data extracted included patient demographics, tumor characteristics, treatment outcomes, adverse events, and study limitations. We did not enforce any specific time period in our search, although the majority of selected articles represent novel research in the spirit of relevance. Inclusion criteria included articles that discussed immunotherapy or radiotherapy individually or in tandem for NSCLC of all stages, as well as surgical resection with either immunotherapy or radiotherapy. Exclusion criteria included articles that were not available in full text, studies published in languages other than English, and grey literature, including conference abstracts, dissertations, and non-peer reviewed articles.

            While not a systematic review, the structure of our informal literature search represents our best attempt to include as many publications relevant to novel NSCLC treatment as possible. Our pointed selection of articles represent a targeted and effective amalgam that presents trusted, standard-setting, and novel research across several research types, to include clinical and laboratorial research as well as systematic reviews and meta-analyses. This review was structured in a manner consistent with the Scale for the Assessment of Narrative Review Articles (SANRA)16.”

  1. 2. Consider providing a diagram or a table to summarize the key combinatorial therapies discussed in the manuscript. This would help readers quickly grasp the main concepts without getting lost in the details.

We thank the reviewer for suggesting the implementation of a table and/or diagram to help readers better digest this digest manuscript. This outside point of view is imperative in helping our manuscript reach a larger audience and for that we are very grateful. Accordingly, we have implemented Table 1, which summarizes the key points and information of trials and studies that are discussed at greater length in the body of our manuscript. Table 1 can be found on line 508 in our manuscript.

  1. Consider expanding the discussion on the challenges of implementing combinatorial therapies in clinical practice (e.g., financial considerations, patient access, need for standardized protocols). This addition would make your review more relevant to a wider audience, including researchers and policymakers.

            This point that the reviewer makes is an excellent one that did not come to our minds when originally writing this manuscript, yet one that is incredibly important to patients and relevant in today’s medical landscape. As such, we have added a section to our discussion section that addresses some of the barriers that exist between patients affected by NSCLC and receiving combinatorial NSCLC treatment, which is becoming increasingly prominent and important. This discourse can be found on lines 511 – 538:

“While the gold standard for NSCLC treatment continues to evolve, it is becoming increasingly evident that treatment optimization will likely involve the combination of several treatment modalities (Table 1). While the rapidly progressing nature of NSCLC treatment has good implications from a scientific standpoint, this comes at the expense of optimized, widely accepted, and tried-and-true guidelines for clinicians to be equipped with. This lack of cemented guidance and standardized protocols, while an opportunity to test treatment efficacy and optimization in clinical trials, is an important limitation in NSCLC treatment today that affects how the disease is approached and tackled from institution to institution, and around the world.

            Financial barriers must also be considered for multi-therapy regimens, particularly for patients who are poorly are not insured, or live in areas of the world where a greater proportion of treatment cost is the patient’s responsibility. While NSCLC treatments vary in cost, no surgical, radioablative, chemotherapeutic or immunotherapeutic agent is inexpensive. Further, the combination of these agents and procedures—while beneficial medically—can substantially increase the cost of treatment89,90. For patients who are uninsured or who are paying out of pocket, combinatorial treatment strategies may not be feasible despite their efficacy. The root cause of this financial burden stems directly from the rapidly evolving nature of the field and therapies that we so often praise; heavy research and development costs, cutting-edge technologies, and heavy investment by third parties all work to drive up consumer cost, which is another factor clinicians should be considering when discussing treatment strategies with their patients. Another important limitation to combinatorial therapy is that of patient access. There are important geographical disparities vis a vis the availability of specialized medical centers that are able to i) diagnose NSCLC in a timely fashion and ii) have the capability to deliver the complex requisite treatments to combat this rapidly progressive and devastating disease. These disparities exist and must be considered not only on a local level, i.e., state- and nation-wide, but also globally. The resources required to adequately deliver NSCLC treatment, let alone combinatorial treatment, can be few and far between and not immediately accessible to patients in middle- and low-income countries, providing a substantial limitation to the implementation of the science we discuss in this review.”

  1. 4. In the era of precision medicine, biomarkers are all the rage. You may consider including a section on potential biomarkers that could guide patient selection for combinatorial therapies in NSCLC (e.g., PD-L1 expression, tumour mutational burden, specific genetic alterations). I would also suggest expanding your discussion on the potential role of circulating tumour cells and cell-free DNA in guiding combinatorial therapies. Blood-based biomarkers are a hot topic right now.

We thank you for providing this feedback, and agree that implementing this would make our evidence-based arguments more robust. As such, we have added a new section to the manuscript, “Markers of Response” on lines 283 – 295, which is dedicated to addressing this suggestion.

“Some biomarkers have been used to predict the response to ICI, including PD-L1 expression, tumor mutation burden, deficient DNA repair markers, and intestinal microbiota8,55–57. To date, however, there are no markers that predict the response to radiotherapy and the response to combined radiotherapy and ICI.

Absolute lymphocyte count has been suggested as a marker of higher and more durable responses to ICI58. The absolute lymphocyte counts before and after radiotherapy and ICI were associated with PFS and the occurrence of the abscopal effect59,60. These findings were corroborated by the demonstration of the association between the occurrence of the abscopal effect and the absolute lymphocyte counts before radiotherapy59. The absolute lymphocyte count after radiotherapy was ten times higher in patients that demonstrated an abscopal effect. 

The mechanistic role of the DAMP leading to an increase in the antigen presentation after radiotherapy induced cell death suggests that DAMP can be used as markers of the increased immunogenicity after radiotherapy. Calreticulin is one of the most widely studied DAMP61,62

Additionally, on lines 381 – 388, we discuss the findings of the circulating DNA in the NADIM trial to make relevant this point in our Surgical Resection for NSCLC section.

“There are no reliable markers that help guide the selection of patients who would benefit from induction with ICI. The radiographic staging before surgery is not reliable, as demonstrated in the NADIM trial. Of all patients with PCR, 33% and 37% demonstrated stable or partial radiographic response after resection, which indicates that radiographic findings do not correlate with pathologic responses. The NADIM trial obtained blood samples and analysis of circulating DNA before resection. In this trial, positive circulating DNA after induction with ICI was associated with a poorer PFS and OS, but such findings needs further validation.”

  1. 5. In section 4 (Surgical Resection for NSCLC), consider discussing the percentage of patients who did not proceed to surgery after neoadjuvant immunotherapy and the intraoperative complication rates. It's also worth considering whether the impressive surgical outcomes reported in relevant randomized trials can be replicated in lower-volume centres. What are your thoughts on this?

Thank you for this point—we agree that discussing patients who do not elect to go to surgery would strengthen this discussion. Accordingly, we added a paragraph in the Surgical Resection for NSCLC section of our manuscript starting on line 374 through 399, we discuss this at length.

“In these trials, a higher percentage of patients in the group that received ICI underwent definitive surgical resection in comparison with patients who received chemotherapy. However, in the CheckMate-816 and the NEOSTAR trials, the main reason for cancellation of surgery was progression of disease from an operable to an inoperable stage followed by an important percentage of patients who developed significant side effects that precluded them from undergoing surgery. Additionally, the benefit of survival was driven by the antitumor effects in patients with advanced tumors (stage III) as opposed to patients with earlier stage tumors, which highlights the need to determine which patients would benefit the most from this approach. There are no reliable markers that help guide the selection of patients who would benefit from induction with ICI. The radiographic staging before surgery is not reliable, as demonstrated in the NADIM trial. Of all patients with PCR, 33% and 37% demonstrated stable or partial radiographic response after resection, which indicates that radiographic findings do not correlate with pathologic responses. The NADIM trial obtained blood samples and analysis of circulating DNA before resection. In this trial, positive circulating DNA after induction with ICI was associated with a poorer PFS and OS, but such findings needs further validation. 

For the purpose of this article, there are no trials to date that compare the role of ICI and chemotherapy with and without RT before surgical resection of NSCLC.  However, a single center study at Cornell evaluated ICI with nivolumab versus nivolumab and SBRT before resection of NSCLC. In 60 patients with resectable stage I-IIIA NSCLC, a randomized phase II study (N = 30 each arm) evaluated the addition of two cycles of neoadjuvant durvalumab to stereotactic body radiotherapy (SBRT) versus neoadjuvant durvalumab alone. The experimental regimen was well tolerated compared to the control and demonstrated a higher MPR rate compared to durvalumab alone (53.3% vs 6.7%) with a crude odds ratio of 16.0 (p < 0.0001). Fifty percent of the tumors that exhibit MPR had a PCR. 74 These results are encouraging and support the synergistic effect of ICI and SBRT without chemotherapy; however, validation at a multi-institutional level and long-term survival data are still needed.”

  1. You've done a great job discussing the clinical evidence for combinatorial therapies in NSCLC, but I think you can take it a step further. Can you provide more detail on the underlying biological mechanisms that make these combinatorial therapies work so well together? For example, what specific immune cell populations and molecular pathways are involved in creating these synergistic effects?

            Thank you for this suggestion. In line with this, we further discuss the biological foundations that underlay NSCLC combinatorial therapies in a new section titled “The Mechanistic approach of radiation-enhanced anti-tumor effects of ICI” on line 242 extending through line 280.

“Recent developments in our understanding of radiotherapy have elucidated possible immunomodulatory mechanisms by which radiotherapy functions beyond the cellular death and senescence caused by direct treatments. The abscopal effect describes the vaccine-like properties that localized radiotherapeutic treatments have purported to have on the systemic immune response to tumors beyond the lesion ablated.39 Scientific evidence, however, has produced contradictory results in terms of the ideal type or dose of radiation. These anti-tumor mediated immune effects of radiotherapy are more frequently associated with low doses40,41.

            Unlike the tumoricidal dose of radiotherapy, low-dose radiotherapy can reactivate the tumor microenvironment by mobilizing innate and adaptive immunity42,43. In contrast, high doses of radiation may induce immunosuppressed effects in the tumor microenvironment mediated by Treg cells and tumor- associated macrophages. However, there is conflicting experimental data related to the effects of low and high dose radiation in the cellular composition of the tumor microenvironment. The myeloid-derived suppressor cells (MDSC) are known to have a tumor-protective effect and promote tumor progression. High doses of radiation induce recruitment and infiltration of tumors and by MDSCs as well as an increase in the number of MDSC infiltrating the spleen, lung, and peripheral blood44. Other studies have demonstrated a reduction in the number of MDSC 1 to 2 weeks after high dose radiation of the primary tumor45. Similarly, dendritic cells (DCs) accumulation and function can be affected by radiation46. Both immunosuppressor and immunostimulatory effects of both high and low doses of radiation have been demonstrated in experimental cancer models. 

It is well accepted that the in situ vaccination effect induced by radiation is the key mechanism inducing systemic effects after local radiation of the tumor. The death of tumor cells after radiation releases multiple neoantigens and damage associated molecular patterns (DAMPs) leading to increased antigen presentation through DCs with subsequent DCs maturation and activation and trafficking to tumors of T cells and NKs cells that destroy the tumor tissues47–49. The understanding of the molecular systemic response to radiotherapy is less clear. An increase of TNFa, IL-1, IL-6, IL-8 at the site of primary radiation has been demonstrated50. However, the systemic and abscopal molecular changes in distant tumors are unknown.

Additionally, low dose radiation can directly induce DNA damage in the immune cells that triggers a danger signal that induces an inflammatory response as opposed to the lethal effect that a high dose of radiotherapy has on the immune cells46. Further research on this topic is needed given the variable degrees of sensitivities to radiation in the different populations of immune cells.  

In experimental murine models for NSCLC, low-dose radiation increased macrophage polarization toward M1 and enhanced the natural killer cells with consequential anti-tumor effects51,52. Furthermore, combinations of low-dose radiation and ICI have demonstrated increased immune cell infiltration of tumors and tumor control40,43,53,54.

In conclusion, low-dose radiation has an immunomodulatory effect that can potentially enhance the anti-tumor effects of ICI.”

  1. Have you thought about including a section on quality of life and patient-reported outcomes? It's important to consider how these combinatorial therapies affect patients beyond just their clinical outcomes. Are these approaches harder for patients to tolerate compared to conventional therapy? What do NSCLC patients actually prefer?

            We thank the reviewer for this consideration. We have indeed considered adding a section on quality of life and patient-reported outcomes, but did not execute in our first submitted manuscript. We appreciate the reviewer bringing this suggestion to light as it gives us the confirmation we need to implement this in our revised manuscript. We address this now in our discussion section, on lines 412 – 427.

“Since the introduction of ICI as part of the treatment for NSCLC, the overall survival has improved. Despite these advances, however, overall survival is still poor. The majority of patients with NSCLC are older and have multiple morbid conditions in addition to cancer, which makes them particularly sensitive to the side effects of therapies. For this reason, the patient-reported outcomes are an important measurement that should be considered in addition to the survival and pathologic response to therapy. The three-year assessment of patient reported outcomes of the CheckMate 001 trial demonstrated that patients who received ICI in the form of nivolumab and ipilimumab had improved global quality of health over time after 30 weeks of treatment, and their mean scores were similar to the general population. Also, patients who received ICI had improved cancer-related symptoms in comparison to chemotherapy75. These findings have been reported in other trials of ICI with or without chemotherapy17,76,77. There are a few areas that need improvement in the field of patient reported outcomes, such as the standardization of the tools used to measure the patient reported outcomes and the validation of these tools to assess the repercussions of ICI toxicities. Effects of the combination of multiple therapies in the patient reported outcomes has not been well established particularly related to the effects of radiation or surgery in addition to ICI.”

  1. Consider discussing in detail the limitations of your review. It's important to be transparent, both in terms of the inherent limitations of a narrative review and any specific limitations related to your manuscript (e.g., quality of included studies).

            We thank the reviewer for suggesting the addition of a section that addresses the limitations of our review. While the limitations of a narrative review, for which there are several, are often implied, we believe that the addition of an explicit limitations section serves our manuscript well and adds to the transparency we build in the aforementioned methods section. The new section addressing the limitations of our review can be found on lines 539 – 562:

“In addition to the scientific, logistic, and pragmatic limitations of combinatorial treatment, our review hosts limitations as well. First, our selection of relevant and impactful studies aimed at portraying the salient findings and impressions of NSCLC treatment is less geared towards holistically describing the state of combinatorial treatment, but more so the important studies that have shaped the field in the past two decades. As such, the inclusion/exclusion criteria for selected articles and studies were not as stringent as those performed in systematic reviews, and not all studies that exist and are available on this topic were included in our discussion. As such, our narrative review is open to several biases that are worth mentioning that may otherwise be mitigated in the systematic review process, including but not limited to selection bias, as aforementioned, publication bias, citation bias, confirmation bias, subjectivity, and lack of reproducibility. We aimed to mitigate some of these biases by having more than one researcher design, perform, and contribute to our search, by including studies that both confirm and refute common hypotheses, and by describing our search criteria—although a narrative and not systematic review—in a methods section.

            Another important limitation of this review is the quality of each study we include in our discussion. This review encompasses a wide berth of various study types, including randomized control trials, observational studies and retrospective reviews, which are held to different standards which might affect the validity of comparing them to each other as we do here. Further, heterogeneities in study design, outcome measures, patient populations, surgical techniques, radiotherapy dosing, fractionation, and scheduling, immune- and chemotherapy dosing, as well as innumerable other metrics make it increasingly difficult to draw parallels and conclusions from one study to another. This limitation highlights the important of clinical trials that directly compare therapy options, such as the CheckMate, NeoCOAST, NEOSTAR, NADIM, and other trials we describe in this review.”

Reviewer 2 Report

Comments and Suggestions for Authors

This article is very comprehensive regarding the modern treatment strategies of lung cancer - NSCLC.

Lung cancer, as the authors pointed out, still remains a problematic issue both for the oncologists and thoracic surgeons, due to low early diagnosis rates and mostly due to the cellular structure of lung tumors, which are mostly heterogenic. Over the past few years it has been proven that NSCLC lung cancer is a disease that is optimally treated in a multimodal fashion, and every step of the way is very important: from diagnosis (clinical, imagistic and pathological diagnosis) to a proper staging, discussion in tumor boards and finally the development of a treatment strategy.

Unfortunately, every step takes some time, which is not favorable for the patients, especially for non-resecable stages (IIIB or higher). Pathologic examination of the biopsy specimen, then immunohistochemy, then bio-markers can take up to 2 months in some countries. 

This is why it is necessary to understand the bigger picture, and this paper tries to do just that. Perhaps o more comprehensive future study that takes into account these combinations of treatments - surgical +- chemotherapy, RT, SBRT or immunotherapy more focused on the staging of the disease - would be even more useful. It is clear that the future of NSCLC treatment is a combination of surgery/SBRT with a more individualized chemotherapy/immunotherapy and RT could bring a more prolonged survival of lung cancer patients.

Author Response

Reviewer 2:

  1. This article is very comprehensive regarding the modern treatment strategies of lung cancer - NSCLC.

Lung cancer, as the authors pointed out, still remains a problematic issue both for the oncologists and thoracic surgeons, due to low early diagnosis rates and mostly due to the cellular structure of lung tumors, which are mostly heterogenic. Over the past few years it has been proven that NSCLC lung cancer is a disease that is optimally treated in a multimodal fashion, and every step of the way is very important: from diagnosis (clinical, imagistic and pathological diagnosis) to a proper staging, discussion in tumor boards and finally the development of a treatment strategy.

Unfortunately, every step takes some time, which is not favorable for the patients, especially for non-resecable stages (IIIB or higher). Pathologic examination of the biopsy specimen, then immunohistochemy, then bio-markers can take up to 2 months in some countries.

This is why it is necessary to understand the bigger picture, and this paper tries to do just that. Perhaps o more comprehensive future study that takes into account these combinations of treatments - surgical +- chemotherapy, RT, SBRT or immunotherapy more focused on the staging of the disease - would be even more useful. It is clear that the future of NSCLC treatment is a combination of surgery/SBRT with a more individualized chemotherapy/immunotherapy and RT could bring a more prolonged survival of lung cancer patients.

            We thank the reviewer for their thoughts on our review, and absolutely agree with them with regards to the point that future NSCLC treatment studies should consider the combination of surgical, systemic, and radioablative therapies to ascertain what the optimal combination, timing, and dosage looks like to confer the greatest survival benefit. We are grateful for this reviewer’s generous review of our manuscript and hope that the reviewer finds our revised manuscript improved.

Round 2

Reviewer 1 Report

Comments and Suggestions for Authors

Thank you for taking the time to revise your manuscript. I can see you have put a lot of work into addressing the feedback from the first round of review. Your changes have significantly improved the rigor and overall quality of the paper.

Overall, I believe your revised manuscript reads much better and makes a valuable contribution to the field. Well done on your hard work.